# Single-Cell RNA Sequencing (scRNA-seq) Identifies L1CAM as a Key Mediator between Epithelial Tuft Cell and Innate Lymphoid Cell in the Colon of *Hnrnp I* Knockout Mice

**DOI:** 10.3390/biomedicines11102734

**Published:** 2023-10-09

**Authors:** Guanying (Bianca) Xu, Yuan-Xiang Pan, Wenyan Mei, Hong Chen

**Affiliations:** 1Department of Food Science and Human Nutrition, College of Agricultural, Consumer, and Environmental Sciences, University of Illinois at Urbana-Champaign, Urbana, IL 61801, USA; gxu8@illinois.edu (G.X.); yxpan@illinois.edu (Y.-X.P.); 2Division of Nutritional Sciences, College of Agricultural, Consumer, and Environmental Sciences, University of Illinois at Urbana-Champaign, Urbana, IL 61801, USA; 3Department of Comparative Biosciences, College of Veterinary Medicine, University of Illinois at Urbana-Champaign, Urbana, IL 61801, USA

**Keywords:** tuft, ILCs, L1CAM, colon, inflammation, colitis

## Abstract

(1) Background: Knockout (KO) of heterogeneous nuclear ribonucleoprotein I (*Hnrnp I*) in mouse intestinal epithelial cells (IECs) induced a severe inflammatory response in the colon, followed by hyperproliferation. This study aimed to investigate the epithelial lineage dynamics and cell–cell communications that underlie inflammation and colitis. (2) Methods: Single cells were isolated from the colons of wildtype (WT) and KO mice and used in scRNA-seq. Whole colons were collected for immunofluorescence staining and cytokine assays. (3) Results: from scRNA-seq, the number of DCLK1 + colonic tuft cells was significantly higher in the *Hnrnp I* KO mice compared to the WT mice. This was confirmed by immunofluorescent staining of DCLK1. The DCLK1 + colonic tuft cells in KO mice developed unique communications with lymphocytes via interactions between surface L1 cell adhesion molecule (L1CAM) and integrins. In the KO mice colons, a significantly elevated level of inflammatory cytokines IL4, IL6, and IL13 were observed, which marks type-2 immune responses directed by group 2 innate lymphoid cells (ILC2s). (4) Conclusions: This study demonstrates one critical cellular function of colonic tuft cells, which facilitates type-2 immune responses by communicating with ILC2s via the L1CAM–integrins interaction. This communication promotes pro-inflammatory signaling pathways in ILC2, leading to the increased secretion of inflammatory cytokines.

## 1. Introduction

The colon is lined with various types of epithelial cells that perform different functions. These cells include absorptive cells and secretory cells—goblet cells, enteroendocrine cells, and tuft cells—with each type of cell performing its own unique functions [1]. Absorptive cells, in general, are responsible for the absorption of water and electrolytes from the colon. Goblet cells secrete mucus, which helps lubricate the colon and protect it from harmful substances. Enteroendocrine cells (EEC) produce hormones that regulate various digestive processes [1]. Tuft cells, interestingly, are a rare type derived from a secretory lineage (they comprise less than 1% of all intestinal epithelial cells) [2], and they take part in chemosensing [3]. Previously published data indicate that tuft cells possess a type-2 helper T cells (Th2)-related gene expression signature [4], and their number increased in the intestine during parasitic infection. Following helminth infection, tuft cells undergo rapid IL4Rα-dependent amplification directed by *Pou2f3* and secrete IL25. Both actions initiate mucosal type 2 immune responses to the parasitic infection [5]. Particularly in mice, tuft-cell-derived IL25 further activates the tissue-resident group 2 innate lymphoid cells (ILC2) at the resting lamina propria in the small intestine after helminth infection. As a result, ILC2s secrete IL13, a cytokine that promotes the self-renewal of intestinal stem cells [6]. Consequently, the differentiation of tuft cells is promoted by epithelial crypt progenitors after receiving IL13 signaling; therefore, it comprises a response circuit between intestinal epithelium remodeling and type 2 immunity [7]. There are a number of intestinal tuft cell markers recognized by previous studies, including DCLK1 for structural support [1], α-Gustducin, GNAT, and TRPM5 for taste functions [8] and SOX9, LGR5 as markers from progenitors [1,9]. Notably, a long-lived, quiescent population of intestinal DCLK1 + tuft cells contributes to recovery following intestinal and colonic injury. Furthermore, they can initiate colon cancer when activated by tissue injury [10]. DCLK1 + tuft cells are also important responders to bacterial-induced colitis by limiting bacterial infiltration into the mucosa through enhanced epithelial repair responses [11].

Heterogeneous nuclear ribonucleoproteins (HNRNPs), also known as polypyrimidine tract binding proteins (PTBs), are a large family of RNA-binding proteins that participate as a splicing factor in many biological processes—such as mRNA stabilization, transcriptional and translational regulations. Therefore, they are vital proteins in the cellular nucleic acid metabolism [12]. HNRNPs function similarly, only differing in terms of domain composition, which contributes to their functional diversity and complexity [13]. At present, 20 major types of HNRNPs have been categorized, including HNRNP I, also known as PTBP1, which functions in the pre-mRNA processing [14] and is associated with miRNA-mediated gene regulation [15]. Many targets of HNRNP I are irregularly spliced in inflammatory and neoplastic intestinal diseases. Previous studies found that HNRNP I not only regulates intestinal epithelium renewal in zebrafish [16,17], but also establishes neonatal immune adaptation and host–microbe homeostasis to prevent colitis and colorectal cancer development in the intestines of mice at the post-transcriptional level upon birth [18].

Loss of HNRNP I function in the colon through the depletion of its encoding gene, *Hnrnp I*, in mouse intestinal epithelial cells (IECs) leads to early onset of spontaneous colitis, followed by the development of colon cancer in both male and female mice [18]. Based on our recent study, both local and systemic inflammation in the IEC-specific *Hnrnp I* knockout (KO) mice were exacerbated by the activation of nuclear factor kappa B (NF-κB) signaling through a mechanism other than the toll-like receptor (TLR) pathway, consequently leading to chronic inflammation in the colon if untreated [19]. Therefore, we believe that this IECs-specific *Hnrnp* KO animal model is ideal for investigating intestinal cell functions and cell–cell communication between epithelial and immune cell lineages during the development of colon inflammation.

In this study, instead of relying on known markers to isolate cell populations from tissues, which may fail to detect rare intestinal epithelial cell lineages, including tuft cells, we used single-cell RNA sequencing (scRNA-seq) technology by 10x Genomics. Cells were collected from the entire colon of IECs-specific *Hnrnp I* KO mice to ensure a complete, unbiased profiling [20] of colonic cell lineages, including epithelial cells, resident immune cells, and submucosal fibroblasts. We identified gene signatures for each major colonic epithelial cell and immune cell lineages between WT and IECs-specific *Hnrnp I* KO mice. More critically, we analyzed cell–cell communications among different epithelial and immune cell lineages, aiming to identify the unique signaling transduction between specific epithelial cells and immune cells in the colon and to delve into the mechanistic understanding of innate lymphoid cells (ILCs)-initiated immune responses during inflammation and the development of colitis. For the first time, we have bridged the existing knowledge gap in colonic immunity by elucidating the intricate interplay between tuft cells and ILCs, mediated by L1CAM, during episodes of colonic inflammation. This not only sheds light on a previously underappreciated communication pathway but also hints at potential therapeutic targets and strategies to manage inflammatory colonic conditions. Furthermore, our findings suggest that such communication might have broader implications in other inflammatory conditions, and even in oncogenesis, underscoring the novel significance of our research.

## 2. Materials and Methods

All authors have access to the study data and have reviewed and approved the final manuscript.

### 2.1. Animals and Diets

All procedures used were described in protocol #20119, approved by the Institutional Animal Care and Use Committee at the University of Illinois at Urbana-Champaign, and were carried out in accordance with the Animal Welfare Act. *Hnrnp I*^flox/flox^; *Villin*^Cre/+^ mice were supplied by Dr. Wenyan Mei’s laboratory (College of Veterinary Medicine, University of Illinois at Urbana-Champaign) and were generated as previously described [18]. The primers used for genotype confirmation PCR were: *Hnrnp I* floxed allele, forward: 5′-CCCATAACTGTCCATAGACC-3′, and reverse: 5′-TGTTGGTAATGCCAGCACAG-3′; *Villin*^Cre/+^ allele, forward: 5′-TGTCCAATTTACTGACCGTACACC-3′, and reverse: 5′-CGCCTGAAGATATAGAAGATAATCG-3′. Only male mice were used in this study and were from the cross of *Hnrnp I*^flox/flox^; *Villin*^Cre/+^ males with *Hnrnp I*^flox/flox^ females. The wild-type (WT) group included the *Hnrnp I*^flox/flox^ mice, and the knockout (KO) group were the *Hnrnp I*^flox/flox^ *Villin*^Cre/+^ mice. The experimental unit for this study was a single animal.

Male WT and KO mice used in the current study (*n* = 14 mice for each genotype) were maintained (group-housed, 2–4 mice per cage) on a 12:12 h light–dark schedule with food and water provided ad libitum. Mice received an AIN93M diet (Research Diets) after weaning (day 21 after birth) for 9 weeks. Specifically, 3 mice from each genotype were utilized for single-cell RNA sequencing (ScRNA-seq), 5 for histology staining (immunofluorescence), and 6 for the tissue cytokine panel assay. The sample size for scRNA-seq was primarily influenced by budgetary considerations, and the sample sizes for other experiments were dependent on the number of animals available at the time the experiment was initiated. While we recognized the importance of achieving optimal statistical power, these logistical limitations guided our final sample size selection.

For ScRNA-seq, littermates were selected to minimize potential confounding factors introduced by genetic or environmental variability. This was an a priori criterion set to ensure consistency and reliability in our findings. For other experiments, the primary inclusion criterion was the age of the animals. Animals with an age difference of no more than one week were selected. Other than age and the littermate criteria for the sequencing experiment, no additional inclusion or exclusion criteria were established. All data points from each experimental group were included in the analysis. There were no exclusions.

### 2.2. Colon Tissue Sampling

At the completion of the feeding period, mice were euthanized by CO_2_ asphyxiation, followed by whole colon removal. Colon tissues from 3 WT and 3 KO littermates were selected for cell isolation; details are described in the next section. Colon tissues from the rest of the animals were first cut open longitudinally, jelly-rolled from the distal to the proximal end, then immersed in liquid nitrogen-chilled isopentane/2-methyl-butane (99%, Thermo Scientific, Cat# 019387.AP, Waltham, MA, USA) for rapid freezing. Each of the frozen colon samples was transferred to a cryo-mold (5 × 15 mm, 5 mm height), right onto few drops of Tissue-Tek OCT compound (VWR, Radnor, PA, USA) at the central bottom of the cryo-mold. After making sure the sample was properly oriented, OCT gently filled the cryo-mold until the entire specimen was covered. Cryo-molds were left on dry ice for 15–30 s to allow for the OCT to completely harden for sectioning. Sectioning was performed on a Leica CM3050 S cryostat (Leica Microsystems, Inc., Deerfield, IL, USA) at −18 °C with 10 µm section thickness.

### 2.3. Single-Cell Sample Preparation

Immediately after euthanasia and necropsy, whole colon tissues from WT and KO male mice (littermates, 3-month-old, male, *n* = 3) were washed twice with ice-cold PBS to remove fecal matter. Tissues were then opened by cutting longitudinally through the lumen, and excess fat was removed during the process. Opened colon tissues were washed again in a clean petri dish containing ice-cold PBS and lightly swirled with forceps to remove any remaining fecal matter. The cell isolation protocol was based on the manuscript published by Gracz et al. [21], with the modifications described below:Epithelial cells: After washing, tissues were digested in ice-cold dissociation reagent #1 [47 mL PBS + 3 mL 0.5 M EDTA (Sigma, Livonia, MI, USA, #E9884) + 75 µL DTT (Sigma, #D0632)] for 20 min. Then, tissues were transferred to dissociation reagent #2 [47 mL PBS + 3 mL 0.5 M EDTA] for 10 min at 37 °C. After incubation, tissues with reagent were shaken for 30 s to release epithelium from the basement membrane, and remnant tissues consisting of submucosa and muscularis were removed for lamina propria digestion. The cell solution was centrifuged at 1000× *g* for 5 min at 4 °C to pellet the cells. Cell pellets were washed with 10 mL PBS containing 10% FBS (Sigma, Livonia, MI, USA, #F0926). The cell solution was then centrifuged again, and the pellet was resuspended in 10 mL HBSS (Sigma, Livonia, MI, USA, #H6648) containing 8 mg Dispase (Sigma, Livonia, MI, USA, #D4693) and incubated for 10 min at 37 °C in a water bath. After incubation, the cell suspension was filtered through 70 and 40 µm nylon cell strainers (Falcon, #352350 and #352340) to exclude large clumps of cells. The filtered cell solution was pelleted again, washed with 10 mL HBSS containing 10% FBS, and then pelleted again. Finally, cells were resuspended in 1 mL RPMI 1640 (Corning Life Sciences, Corning, NY, USA, #15-040-CV) with 10% FBS.Lamina propria cells: Remnant tissues from previous steps were transferred to digestion media (5 mL RPMI 1640 + 2 mg Dispase + 10 mg collagenase IV (Gibco, Grand Island, NY, USA, #17104-109) + 60 µL FBS) and cut into small pieces, incubated for 30 min at 37 °C with constant spinning. After incubation, the cell suspension was filtered through 70 and 40 um nylon cell strainers to exclude large clumps of cells, the same manufacturer as described before. The filtered cell suspension was pelleted again, washed with 10 mL HBSS containing 10% FBS, then pelleted again. Finally, cells were resuspended in 1 mL RPMI 1640 with 10% FBS.

Cell suspension from the epithelium and lamina propria were combined and subjected to dead cell removal following the manufacturer’s instructions (Miltenyi Biotec, San Diego, CA, USA, #130-090-101 and #130-042-201). The cell suspension after the dead cell removal is considered whole-colon cells and is ready for sequencing.

### 2.4. cDNA Library Construction and Single-Cell RNA-seq

Cellular suspensions (~5000 cells) were loaded on a Chromium Single-Cell Instrument (10× Genomics) to generate single-cell GEMs. Single-cell RNAseq libraries were prepared using Chromium Single-Cell Gene Expression NextGem V3.1 and sequenced on an S4 2 × 150 nt lane in a NovaSeq 6000 platform, following the manufacturer’s protocol (10× Genomics). Reading files (150 nt in length) in FASTQ format were generated and demultiplexed by Cell Ranger 3.1.0 (10× Genomics Cell Ranger 3.0.0) aligned with mm10 mouse reference genome. Sequencing and library construction were performed at the Roy. J. Carver Biotechnology Center, University of Illinois at Urbana-Champaign.

### 2.5. Process and Quality Control of the Single-Cell RNA-seq Data

The exonic reads uniquely mapped to the transcriptome were used for a unique molecular identifier (UMI) counting. Subsequently, single cells and their UMI count matrices were imported into the R package “Seurat” (version 2.3.2) for further analysis [22]. Selection and filtering of the single-cell RNA sequencing data were carried out if the total UMI counts (cell counts) or the number of expressed genes (feature counts) of cells were beyond predefined thresholds, defined as the medians ± 3× the median absolute deviation (MAD). We also discarded both the genes expressed in fewer than three cells and low-quality cells with ≤200 expressed genes. Furthermore, cells with >25% of the mitochondrial genes (markers for dead cells) were also discarded. After the filtering process, as described above, a total of 31,798 cells collected from 6 animals were kept for subsequent analysis.

### 2.6. Single-Cell RNAseq Data Analyses

Single-cell RNAseq data analyses were performed using Cell Ranger (10x Genomics Cell Ranger 3.0.0), R software (4.0.5; 2021-03-31), and Loupe Browser (10x Genomics Loupe Browser 5.1.0). Gene expression was statistically processed using Cell Ranger with the negative binomial exact test based on a published method [23]. Scale normalization of the processed gene expression data was carried out with Seurat 4.0.3 [24], the SCTransform method, following standard Principal Components Analysis (PCA), which reduced 25 clusters to two dimensions using the uniform manifold approximation and projection (UMAP), a dimension reduction technique. A computational cell calling algorithm (Seurat; SingleR, ver 1.7.1) was used to identify the cell type of each cluster by comparing their overall expression patterns with two published annotated mouse cell data sets. Cell lineages of different clusters of the same cell type (i.e., epithelial cells, T cells and B cells) were assigned by filtering overall expression in cells from different clusters with established, unique cell markers. UMAP figures with assigned cell types for each cluster were generated.

Proportions of epithelial cell lineages including tuft, goblet, Paneth, intestinal stem cells were calculated using the following formula:$$\frac{total\;number\;of\;cell\;within\;the\;lineage\;clsuter\;collected}{total\;number\;of\;epithelial\;cells\;collected\;\times \;colon\;length\left(cm\right)}\times 100\%$$

Immune cell subtypes’ (ILCs, NK, T cells, B cells, myeloid progenitors) proportions were calculated using the following formula:$$\frac{total\;number\;of\;immune\;cells\;subtype\;collected}{colon\;length\;(cm)}.$$

Aggregated Seurat object was exported to a UMAP metadata file and imported to Loupe Browser 5.1.0 to calculate and make comparisons. In addition, a CellChat (ver 1.1.3) [25] object was also created from the Seurat object to perform cell–cell communication analysis, using all default settings following the publisher’s vignette.

### 2.7. Immunofluorescent (IF) Staining

Sections used for IF staining were blocked with a blocking buffer (1× PBS/1% bovine serum albumin/5% normal goat serum/0.05% Triton X-100). Incubations were performed following the respective procedures. In brief, primary antibodies of anti-DCAMKL1/DCLK1 (Abcam, Cambridge, UK, ab31704, dilution at 1:200), anti-L1CAM (Abcam, Cambridge, UK, ab24345, dilution at 1:100), anti-GATA3 (Abcam, ab199428, dilution at 1:200) or anti-ST2 (Abcam, Cambridge, UK, ab25877, dilution at 1:300) were diluted in the antibody dilution buffer and incubated with the sections in the dark and overnight at 4 °C; after the primary incubation, sections were washed with 1× PBS and incubated with goat–anti-rabbit IgG-Alexa Fluor^®^ 488 (Cell Signaling Technology, Dambers, MA, USA, CST#4412, dilution at 1:200) or goat–anti-rabbit IgG-Alexa Fluor^®^ 647 Conjugate secondary antibody (Invitrogen by Thermo Fisher, Waltham, MA, USA, Cat # A-21245, dilution at 1:200) at room temperature for one hour; after primary and secondary incubation, all slides were washed in 1× PBS and incubated with DAPI to stain nucleus (Invitrogen by Thermo Fisher, Waltham, MA, USA, Cat # D1306, diluted to1 µg/mL) for 5 min in the dark at room temperature. Sections were then washed with 1× PBS, mounted with Prolong-Gold Antifade Reagent (Molecular Probes by Life Technologies, Carlsbad, CA, USA), then dried in the dark overnight at 4 °C. Images were captured using the Confocal LSM 700 microscope (Carl Zeiss Microscopy, LLC., White Plains, NY, USA) with an objective lens that has a magnification power of 20 times (20×) or 40 times (40×), and acquired using Zen software (Carl Zeiss AG, White Plains, NY, USA) and analyzed with using Image J software (U. S. National Institutes of Health, http://imagej.nih.gov/ij/ (accessed on 1 March 2021)). Tuft cell numbers in each colon sample were counted as ratio of positively stained cells to the number of crypts within a fixed 800 × 800 µm captured area. Results are presented as the mean positive cell/crypt ± SEM.

### 2.8. Colon Tissue Protein Extraction and Multiplex Cytokine Analysis

Total protein was extracted from frozen colon jellyrolls using a protocol adapted from Boster Bio (bosterbio.com (accessed on 1 November 2021)). In brief, ~3–5 mm sections of frozen colon jellyrolls were crosscut using a surgical blade and washed in 1 mL ice-cold 1× PBS buffer. The colon tissue was then suspended in the protein extraction buffer (PEB, 100 mM Tris-HCl, pH 7.5, 150 mM NaCl, 1 mM EGTA, 1 mM EDTA, 1% Triton X-100, and 0.5% Sodium deoxycholate) containing 1 mM PMSF, 1× protease inhibitor (Roche, Basel, Switzerland), 1× phosphatase inhibitor 2, and 1× phosphatase inhibitor 3 (Sigma, St. Louis, MO, USA), and sonicated using a sonic dismembrator (Fisher Scientific, Waltham, MA, USA). Protein concentrations were determined using the long Lowry Assay as previously described [26,27]. Colon tissue levels of cytokines were assessed using the multiplex cytokine assay Mouse Cytokine 31-plex Discovery Assay Array by Eve Technologies (Eve Technologies Co., Calgary, AB, Canada).

### 2.9. Statistical Methods

Comparisons of cell lineage proportions and DCLK1 + stained tuft cell per crypt between WT and KO mice were analyzed via unpaired parametric *t*-test in GraphPad Prism 9.4.1. Cytokine expression levels in the colon between WT and KO mice were compared using Mann–Whitney non-parametric *t*-test (GraphPad Prism 9.4.1). Data are presented as means ± SEM, with *p*-values < 0.05 considered significant. Gene expression levels in cells were analyzed via negative binomial exact test based on the published sSeq method [28] in Loupe Browser 5.1.0, generating adjusted *p*-values for multiple testing using the Benjamin-Hochberg procedure to control false discovery rate (FDR). Data are presented in violin plots, with *p*-values < 0.1 considered significant.

## 3. Results

### 3.1. RNA-seq Suggests Significant Increases of Tuft Cell and ILC Lineages in Response to the Depletion of Hnrnp I in the Colon Epithelia

A total of 18 cell lineages were identified. Unique cell markers used to identify each cell lineage are listed as follows: absorptive colonocytes (Car1, Ceacam1), stem cells (Mki67, Slc12a2), goblet cells (Muc2, Fcgbp), endothelial cells (Ccl21a, Cdh5), enteroendocrine cells—EECs (Chga, Chgb), Paneth cells (Reg4, Mptx1), tuft cells (Dclk1, Lrmp), fibroblast (Col1a1, Col3a1, Dcn), CD45^+^ lymphocytes (Cd8a, Ptprc, Cd69), cytotoxic T cells (Nkg7, Ccl5), non-cytotoxic T cells (Cd4, Il7r), B cells (memory and naïve) (Cd79a, Cd79b), B cells (plasma) (Igha), mast cells (Mcpt1, Mcpt2), macrophages (C1qa, C1qb, and C1qc), myeloid progenitors (Cd300c, Cd4, Cd8a), natural killer/NK cells (Cd7, Trbc2, Cd3g, Cd3d), and innate lymphoid cells—ILCs (Ncr1, Ifng, Gata3). Cells expressing mostly hemoglobin genes (Hba−, Hbb−) were considered contaminated by red blood cells and therefore labeled as erythrocytes. All clusters are shown in Figure 1A. After normalizing cell numbers in each cluster, tuft (*p* = 0.0617) and ILCs (*p* = 0.0363) proportions were increased in the colons in KO mice compared to WT (Figure 1B). The results indicate an increased need for these cell lineages in KO mice colon, and a subsequently higher possibility of communication between the tuft–ILC circuit responding to the inflammation caused by the ablation of Hnrnp I.

To confirm whether the increase in colonic tuft cells in Hnrnp I KO mice is significant, as indicated by the scRNA-sequencing result, colon sections were stained with anti-DCLK1, an established tuft cell marker, as shown in Figure 2A,B. The quantification of tuft cells was described in the method; results are shown in Figure 2C. In KO mice, the DCLK1^+^ tuft cell is 0.89 ± 0.55 per crypt, and in WT this is 0.55 ± 0.04 per crypt. KO mice had significantly increased DCLK1^+^ tuft cells along the colonic crypts, which confirms the scRNA-seq results that KO mice have more developed tuft cells along the colon compared to the WT mice.

### 3.2. ScRNA-seq Data Show Hnrnp I KO-Specific Cellular Communications from Colon Tuft Cells to Lymphocytes via the L1 Cell Adhesion Molecule (L1cam) Signaling, Particularly to ILCs, by Interacting with Integrin Subunits av + b3

Because the number of colonic tuft cells increased in Hnrnp I KO mice, we hypothesized that the cellular communications from tuft cells to immune cells, such as the number of and strength of interactions, were also affected. Therefore, we investigated the possible interactions between tuft cells and other immune cell types [NK cells, CD45^+^ lymphocytes, cytotoxic T cells, stem cells (intestinal), helper T cells, ILCs, and myeloid progenitors] using CellChat, an R package exploring the ligand–receptor interactions. We found that communications from tuft cells to other immune cells via L1cam are only possible in KO mice (*p*-value < 0.01). Although the L1cam-directed communication possibilities are low, as indicated by the spectrum, this is specific to tuft cells in KO, meaning it is not detected between tuft cells and any other lymphocytes in WT. Results are shown in Figure 3A. ScRNA-seq results indicate that KO mice tend to have more tuft cells expressing a higher level of L1cam (*p*-value = 0.076), as shown in Figure 3B, which could explain the specificity of the interaction in KO. Interestingly, we found that all lymphocytes can communicate with tuft cells through interactions between L1cam and integrin subunits α4 + β7. However, the interaction between L1cam and integrin subunits αv + β3 is unique and only possible from tuft cells to ILCs (Figure 3A). ScRNA-seq results indicate that KO mice have more ILCs expressing higher levels of integrin subunits αv (*p*-value = 0.89) and β3 (*p*-value = 0.15), while the expression levels of integrin subunits α4 (*p*-value = 0.9) and β7 (*p*-value = 0.11) are the same among different lymphocyte clusters (Figure 3C). This could explain the ILC specificity of interactions between L1cam and integrin subunits αv + β3 in our colon inflammation model.

To confirm the protein expression level of L1CAM on the tuft cell surface, we incubated the whole colon sections with anti-L1CAM and anti-DCLK1. Results are presented in Figure 4A,B. Consistent with the scRNA-seq data, we observed high L1CAM expression (green) on DCLK1^+^ the tuft cell surface (red) only in KO mice, suggesting that L1CAM-directed interactions from tuft cells to other cell lineages are specific to the colon inflammation model.

### 3.3. Tuft Cells Activate NF-kB in ILCs via L1cam–Integrins Interactions to Promote ILC2 Differentiation and Autophagy, Consequently Resulting in Increased Inflammatory Cytokines—IL4, IL6, and IL13

Since ILC lineage was significantly increased in the colons of KO mice, and there is strong evidence suggesting interactions between colonic tuft cells and ILCs that are directed by L1cam–integrins interactions, NF-kB signaling in ILCs was investigated to identify the responders and downstream consequences. ScRNA-seq results indicate that in KO animals, ILCs expressed similar levels of Nfkb1, Nfkb2, Rela, and Relb (encoding proteins P50, P52, P65, and RELB, respectively, all of which are NF-kB signaling subunits) compared to WT animals. On the other hand, Rel, which encodes NF-kB subunit c-REL, was the only NF-kB signaling subunit that was significantly increased in the KO ILCs (*p*-value = 0.052, Figure 5). Because c-REL is essential for the activation of group 2 ILCs (ILC2), we next examined the markers of ILC2 to confirm the consequences of L1CAM-integrins activated NF-kB signaling. Kit (encoding c-KIT, which marks a functional subset of ILC2), Rora (encoding RORA, which is required for ILC2 differentiation), Batf (encoding BATF, which is required for ILC2 homeostasis) were found to be upregulated (*p*-value = 0.016, 0.099, and 0.06, respectively) in the ILCs of KO mice, suggesting that ILC2 was the major responder among ILC groups in our colonic inflammation model (Figure 5). In addition, we also found that autophagy regulators Ddit3 (encoding CHOP) and Sqstm1 (encoding P62) demonstrated different expression levels (*p*-value = 0.072 and 0.0058, respectively) between WT and KO ILCs (Figure 5). Autophagy is ultimately utilized by activated ILC2s during their maintenance of homeostasis and effector functions; therefore, the results also indicate that ILC2 was the major responder during inflammation in our colon inflammation model.

A cytokine array was performed to measure the expression levels of inflammatory cytokines in colon tissues. The results suggest that IL4, IL6, and IL13 concentrations in the colon are significantly higher in KO mice compared to WT (*p*-value = 0.0119, 0.0065, and 0.0465, respectively, Figure 6A), while IFNγ, TNFα, IL17, and G-CSF concentrations remain unchanged between KO and WT mice in the colon (Figure 6A). In ILCs, there are mainly three subtypes: ILC1, ILC2 and ILC3. The secretion of IL4, IL6, and IL13 is mostly performed by ILC2, while ILC1 secrete IFNγ and TNFα, and ILC3 secrete IL17 and G-CSF. To confirm the presence of ILC2 in the colons of KO mice, immunofluorescence staining was performed with anti-GATA3, an ILC2 marker. Representative staining results are presented in Figure 6B, with GATA3 stained in green and the nucleus stained in blue. In the colons of WT mice, visually, there was no presence of ILC2 at the lamina propria, while GATA3 + ILC2s were abundantly present in the lamina propria in the colons of KO mice.

Taken together, the communications from tuft cells to ILCs during colon inflammation are targeted to ILC2 in the colon, specifically via L1CAM–integrins in our mouse model. This interaction subsequently triggers the NF-kB signaling in ILC2 and regulates its autophagy, promoting the secretion of pro-inflammatory cytokines such as IL4, IL6, and IL13.

## 4. Discussion

### 4.1. Colon-Inflammation-Specific Cell–Cell Communication between Tuft Cells and Immune Cells through the L1CAM–Integrins Interactions

The current study utilized high-throughput, highly specific single-cell RNA sequencing to map the cell–cell communications in the novel *Hnrnp I* KO mouse model mimicking the colon inflammation microenvironment. We are the first to identify a unique, colon-inflammation-specific communication between tuft cells and immune cell lineages, particularly group 2 ILC (ILC2), through the L1CAM–integrins interaction. Importantly, we discovered that one of the epithelial lineages, the Tuft cell, is increased in the crypts of the *Hnrnp I* KO mice in response to the epithelial-specific induction of the NF-κB pathway and possibly the consequential microbial dysbiosis and inflammation. As a result, the heightened tuft-cell-initiated communication through L1CAM–integrin interactions leads to the activation of ILC2, which is critical in the downstream immune responses that induce and maintain inflammation and increase the potential risk of colitis, even cancer development.

### 4.2. L1CAM: Function and Signaling

L1CAM is a 200–220 kDa transmembrane glycoprotein from the immunoglobulin superfamily that can interact with itself (homophilic interaction) or with heterophilic ligands such as integrins, CD24, neurocan, neuropilin-1, and other members of the neural cell adhesion family [29,30,31]. In human patients with uterine and ovarian carcinomas, or pancreatic ductal adenocarcinoma, aberrant L1CAM expression was observed and accompanied by aggressive phenotype and tumor stages, making L1CAM a potential indicator of poor prognosis [32,33]. The downstream signaling of L1CAM is activated via its adapter molecules with integrins, growth factor receptors, or protein kinases, either directly or indirectly. L1CAM signaling has two additional pathways—“forward” and “reverse” pathways. In “forward” L1CAM signaling, L1CAM sheds its extracellular domain, then releases its intracellular domain (L1-ICD) to be translocated. The soluble extracellular domain acts as a ligand for integrins and functions during cellular migration, while the L1-ICD regulates L1CAM-dependent genes [34]. In “reverse” L1CAM signaling, full-length, membrane-bound L1CAM interacts with integrins and ezrin without cleavage of L1CAM. This L1CAM pathway leads to the activation of the constitutive nuclear factor NF-κB pathway by enhancing the IL1β expression [35,36,37], which further promotes cell motility, tumor cell growth, and cancer metastasis [38,39,40]. This signaling pathway was observed in human colorectal cancer cells as well as in human pancreatic adenocarcinoma cells [40,41]. In our study, we observed possible communications between colonic tuft cells and lymphocytes via the L1CAM–integrin interaction, which is highly likely through the “reverse” L1CAM pathway and, therefore, crucial for the L1CAM-mediated activation of the NF-κB pathway.

### 4.3. Activation of the NF-κB Pathway in ILCs as a Consenquence of L1CAM-Mediated Communication

Following the L1CAM–integrins interaction, the NF-κB pathway is activated in ILCs. This process is accompanied by the upregulation of *c-Rel* mRNA expression—a dimer subunit and transcription factor in this pathway. Interestingly, *c-Rel* is also a critical mediator of type-2 immune response due to its modulation, particularly ILC2 effector functions [42]. ILC2s belong to the ILC lineage and are mostly found at mucosal tissue sites, including both the small and large intestine [43,44]. ILC2s are activated by inflammatory cytokines such as IL25 and IL33 [45,46,47,48,49,50,51], which drive ILC2-associated mucosal inflammation. Therefore, ILC2 is considered a major component of type 2 immunity in mice, and is believed to be crucial in intestinal immunity against infections [52]. When the NF-κB pathway is activated in ILC2, it stimulates ILC2 to secrete inflammatory cytokines such as IL5 and IL13. These cytokines will facilitate ILC2 progenitors to migrate from the bone marrow to peripheral lymphoid organs [50,53]. Therefore, activation of the NF-κB pathway is essential for ILC2 to function locally. In addition to the immune signals, ILC2 can also pick up metabolic signals, such as dietary stress or malnutrition, at tissue sites. For instance, the development of ILC2 is preferred over ILC3s in the small intestine during vitamin A deficiency or deprivation [54].

### 4.4. Molecular Insights into ILC2s during Colonic Inflammation

Following the activation of the *c-Rel*-dependent NF-κB pathway, we observed an upregulation of *c-Kit, Rora, Batf*, *Sqstm1 (p62)*, and downregulation of *Ddit3 (Chop)* in the ILCs. A high expression of *c-Kit* can be found in those ILC2s with a high level of cell plasticity [55] and is essential for inducing type-2 immunity through the production of IL13 [56]. RORA, a member of a protein family functioning as ligand-dependent transcription factors, is a critical checkpoint for ILC2 commitment. It represses T-cell development while promoting ILC2 development in the embryonic thymus [57]. Increased mRNA expression of *Rora* indicates an increased proportion of ILC2s, as observed in the peripheral blood of chronic obstructive pulmonary disease patients [58]. During helminth infection in the lung and intestine, ILC2s produced IL4 and IL13 using a BATF-dependent mechanism in response to epithelial damage and contributed to the epithelial restoration [59]. In addition, BATF depletion may result in defective responses from ILC2 to inflammatory cytokines during bacterial infection [60]. Therefore, BATF was recognized as an essential regulator for ILC2 function. Taken together, the upregulation of *c-Kit, Rora, and Batf* indicates that a group of highly proliferative ILC2 with high plasticity was present in the colons of KO animals secreting inflammatory cytokines. We also noticed that autophagy was extensively manipulated in the ILCs in KO mice. P62, encoded by *Sqstm1*, is a classical receptor of autophagy, functioning during the proteasomal degradation of ubiquitinated proteins [61]. The inhibition of autophagy leads to P62 accumulation in cells [62]. C/EBP homologous protein (CHOP), encoded by *Ddit3*, has also been linked with autophagy [63] and the regulation of cell death, as well as cell survival [64]. In our study, we observed restricted autophagy in the ILC cluster. This restriction could be particularly targeted to the cell metabolism in ILC2, reprogramming their fuel dependency from fatty acids to glucose, as well as impairing their cytokine production [65]. Our results indicate that ILC2 was likely the major responder to L1CAM-mediated NF-κB signaling among ILC subtypes. This conclusion was made based on the observation that genes essential to ILC2-activation, maintenance, and effector functions were significantly changed in the KO-ILC cluster. Moreover, we observed that concentrations of ILC2-secreted cytokines (IL4, IL6, and IL13) were significantly increased in the KO mice colon compared to WT. Among these cytokines, IL13 is most effective in patients with ulcerative colitis (UC) and fistulizing Crohn’s disease (CD), which makes anti-IL13 agents a promising therapeutic strategy for the management of inflammatory bowel disease (IBD) [66]. In a murine model with oxazolone-induced colitis, which resembles human UC, IL13 is essential for the induction and development of colitis [67]. Similarly, in our animal model, IL13 could be the inducing factor of colonic inflammation.

### 4.5. L1CAM’s Potential as a Biomarker in Colorectal Pathophysiology

While its promise is evident, translating L1CAM into a diagnostic and prognostic biomarker requires further validation across diverse patient cohorts to ensure its broader applicability and efficacy in managing conditions raised from chronic colonic inflammation, such as colorectal cancer (CRC). In recent years, two non-invasive biomarkers that can reliably detect and prognosticate CRC have been discovered in clinical settings: the Kirsten rat sarcoma viral oncogene homolog (KRAS) and the B-raf serine/threonine kinase proto-oncogene (BRAF) [68,69]. Both KRAS and BRAF mutations have been associated with colorectal tumorigenesis and serve as critical indicators for therapeutic strategies and disease prognosis, providing valuable insights into potential therapeutic targets. In this context, we propose evaluating L1CAM as another potential biomarker, not directly for CRC detection, but rather as an indicator for the initiation of immune responses, a factor that might play a pivotal role in the broader landscape of colonic inflammation and subsequent oncogenesis. The rationale behind this is that L1CAM’s potential role in mediating cell–cell communications, as indicated by our recent findings, might offer unique insights distinct from the molecular pathways in which KRAS and BRAF are involved. Thus, L1CAM could potentially complement the information provided by KRAS and BRAF, offering a more holistic view of the colorectal cellular environment, from oncogenesis to immune modulation.

### 4.6. Study Limitations and Considerations for Future Research

Lastly, we would like to discuss several limitations that must be considered when interpreting the findings of this current study. Firstly, we did not perform in vitro experiments using colon organoids to establish a direct causal relationship between L1CAM and the subsequent inflammatory cascade. Our data, in its current form, elucidate a correlation, hinting at a potentially deeper interplay that warrants further exploration. To provide a comprehensive understanding, it is imperative to undertake additional experimental studies that specifically probe the intricacies of this interaction. Ideally, we would have inhibited the L1CAM mediated cell–cell communication, via drugs or Si-RNA, using in vitro experiments (i.e., colon organoid) to confirm its direct role in modulating the immune responses initiated by ILCs. The use of colon organoids would have allowed us to recapitulate the in vivo environment in a controlled setting, providing a platform that could be used to directly observe and quantify the effects of L1CAM inhibition on the activation of ILC2 cells. The absence of these experiments means that we cannot definitively establish a causal link between L1CAM-mediated cell–cell communication and immune responses, although our findings strongly suggest this relationship. Secondly, our sample size was relatively limited, which impacted the statistical power of our findings. Larger and more diverse sample sizes would be necessary to robustly ascertain the translational potential of our results. Future research should consider integrating in vitro assays and expanding the study population to include different models and larger sample sizes.

## 5. Conclusions

The current study demonstrates one key cellular communication between colonic tuft cells and ILCs, particularly to ILC2, through the interactions between L1CAM and integrin subunits (αv and β3) in colonic inflammation (summarized in Figure 7). This communication potentially activates the *c-Rel*-dependent NF-κB signaling in ILC2, promoting its activation and effector functions, simultaneously initiating type 2 immune responses by increasing the secretion of inflammatory cytokines—IL4, IL6, and IL13. As a result, the elevated production of IL13 further contributes to the induction and development of colitis in the IECs-specific *Hnrnp I* KO mice. Overall, we believe that the discovery of the L1CAM-mediated communication pathway offers promising therapeutic avenues for managing downstream diseases precipitated by both acute and chronic colonic inflammation. Potential interventions targeting the c-Rel-dependent NF-κB signaling in ILC2 could provide novel means to control the aberrant inflammatory responses underlying these conditions. Beyond therapeutic implications, the highlighted cellular communications and resultant cytokine profiles could inspire the development of new diagnostic tools, allowing for earlier detection and possibly paving the way for more personalized treatment approaches for patients affected by colon inflammation.

## Figures and Tables

**Figure 1 biomedicines-11-02734-f001:**
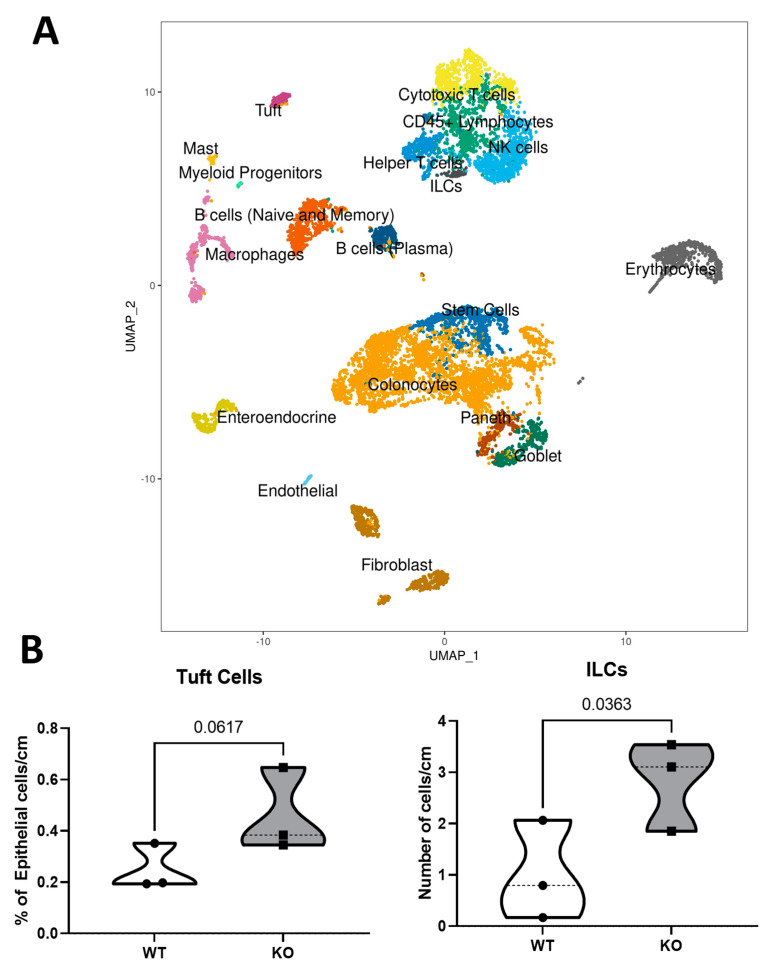
Single-cell RNA sequencing identified cell clusters and subtype proportions. (**A**): UMAP is used to visualize the clustering (color coding) of 31,798 single cells extracted from the mouse colon (*n* = 3 mice per WT or KO) based on the expression of known marker genes. ILCs, innate lymphoid cells; NK cells, natural killer cells. (**B**): Cell proportions of epithelial cell subtypes—tuft and immune cell subtypes—ILCs. Statistical analyses are presented as *p*-values (*n* = 3 mice).

**Figure 2 biomedicines-11-02734-f002:**
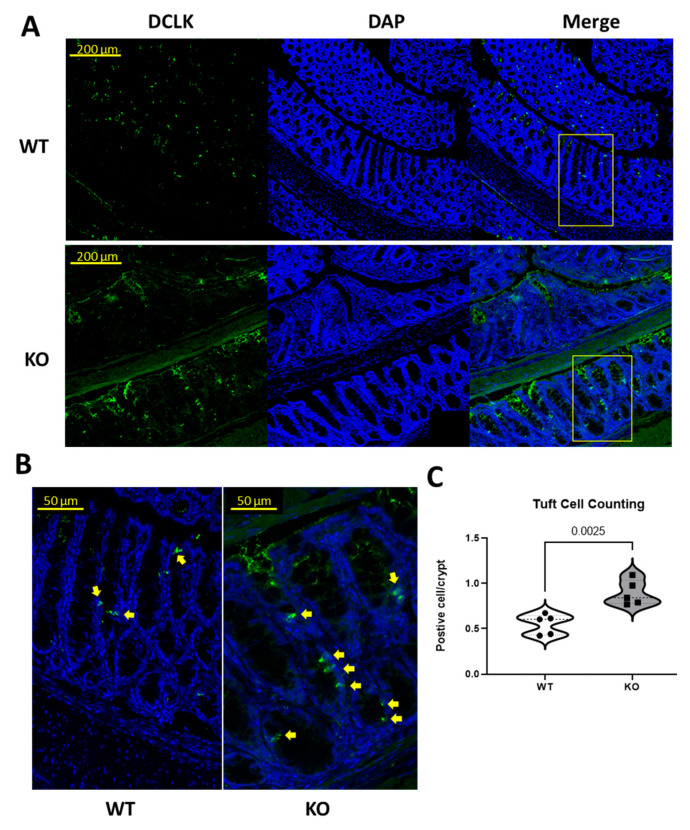
Representative immunofluorescence staining for DCLK1 (tuft cell marker) and quantification analysis. (**A**): Tile scan of representative immunofluorescence staining for DCLK1 (tuft cell marker, green) and merged images (with DAPI, nuclear staining, blue) on WT and KO mouse colon (jelly-rolled). Scale bars = 200 µm. (**B**): Close-up of representative immunofluorescence staining for merged images (DCLK1 and with DAPI) on WT and KO mouse colon (jelly-rolled) used for quantification purposes. Scale bars = 50 µm. (**C**): Quantification of tuft cells on WT and KO mouse colon measured as DCLK1-positive cells/crypt, presented in violin plot. Each dot represents one animal, and the *p*-value is labelled (*n* = 5 mice).

**Figure 3 biomedicines-11-02734-f003:**
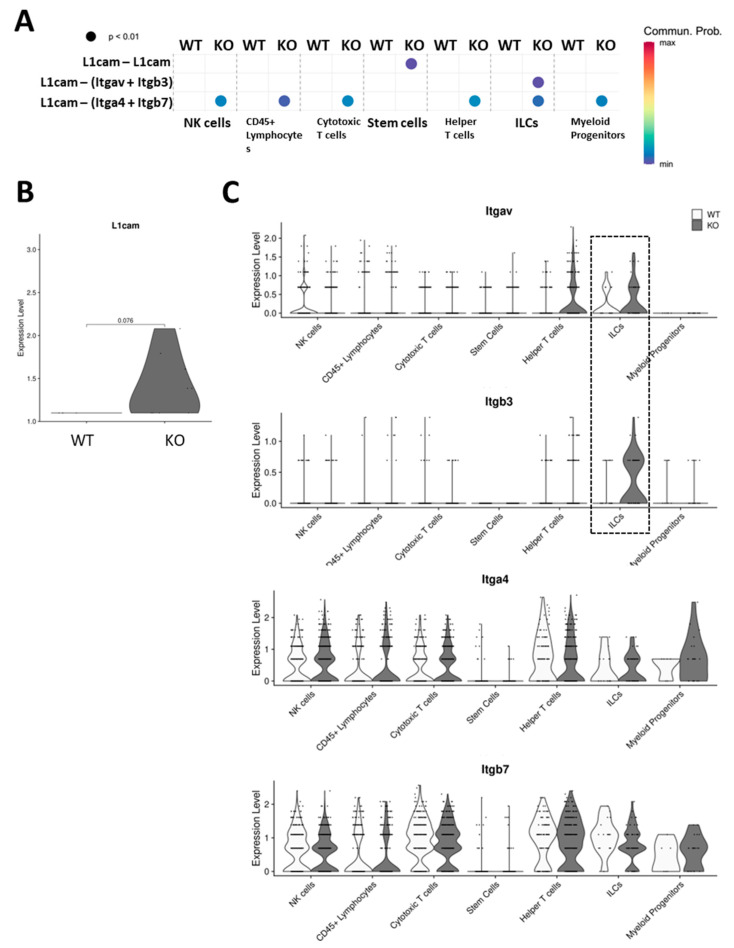
Genetic-specific cell-to-cell communications between colon tuft cells and lymphocyte subtypes and expression levels of the involved ligand receptors. (**A**): Comparison of significant ligand-receptor pairs between WT and KO, which contribute to the signaling from colon tuft cells to lymphocytes subpopulations, including NK cells, CD45^+^ lymphocytes, cytotoxic T cells, stem cells, helper T cells, ILCs, and myeloid progenitors. Dot color reflects communication probabilities, and dot size represents computed *p*-values. Empty space means the communication probability is zero. *p*-values are computed from the one-sided permutation test. Communication via L1cam from tuft cells to lymphocytes is KO-specific (*n* = 3 mice). (**B**): Expression of L1cam in tuft cells on WT and KO mouse colon, presented in violin plot based on scRNA-seq data. Each dot represents a single cell. A difference was detected with a *p*-value = 0.076. (*n* = 3 mice). (**C**): Expression of Itgav, Itgb3, Itga3, and Itgb7 in lymphocyte subpopulations on WT and KO mouse colon, presented in violin plot based on scRNA-seq data. Each dot represents a single cell. A significant difference was not detected. (*n* = 3 mice).

**Figure 4 biomedicines-11-02734-f004:**
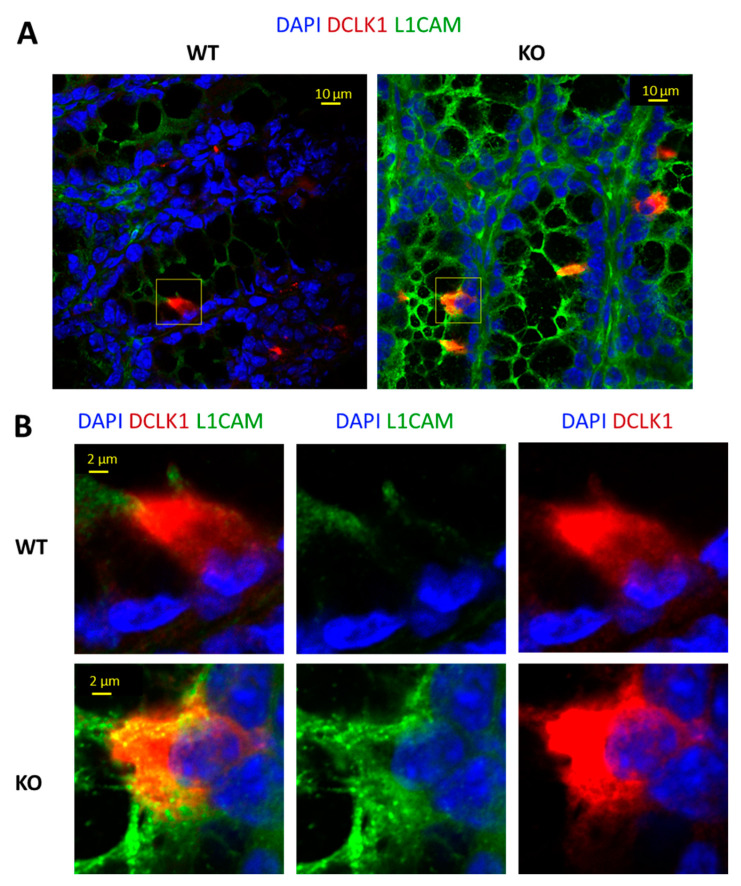
Representative immunofluorescence staining for DCLK1 (tuft cell marker) and L1CAM (signaling ligand). (**A**): Single scan with an objective lens that has a magnification power of 40 times (40×), demonstrating representative immunofluorescence staining for DCLK1 (tuft cell marker, red), L1CAM (ligand, green) merged with DAPI (nuclear staining, blue) on WT and KO mouse colon (jelly-rolled, *n* = 4 mice stained). Scale bars = 10 µm. (**B**): Close up of DCLK1-positive cell (colon tuft cell) expressing a low level of L1CAM (WT) and high level of L1CAM expression (KO) merged with DAPI; L1CAM expression merged with DAPI; DCLK1 merged with DAPI, respectively. Scale bars = 2 µm.

**Figure 5 biomedicines-11-02734-f005:**
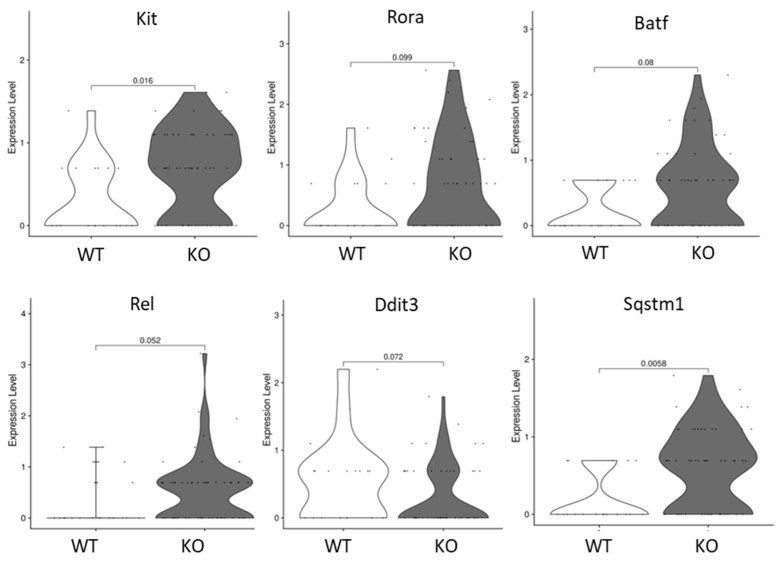
Expression levels of genes involved in NF-κB, ILC2 activation and maintenance, and autophagy. Expression of genes regulating ILC2 activation, maintenance, and effector functions—Kit, Rora, and Batf, NF-κB subunit c-Rel, and genes regulating autophagy—Ddit3 and Sqstm1 in ILCs in WT and KO mouse colon, were presented in violin plot based on scRNA-seq data. Each dot represents a single cell. Significant difference was detected at a *p*-value < 0.05. (*n* = 3 mice).

**Figure 6 biomedicines-11-02734-f006:**
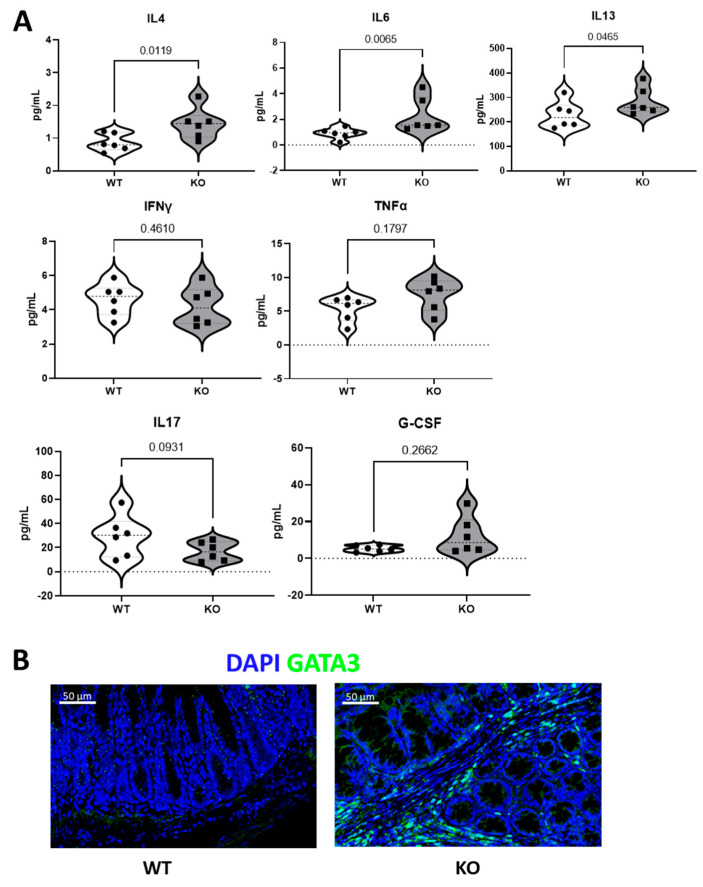
Expression levels of inflammatory cytokines and representative immunofluorescence staining of ILC2 in colonic lamina propria. (**A**): Expression of ILC1-secreted cytokines—IFNγ and TNFα; expression of ILC2-secreted cytokines—IL4, IL6, and IL13; expression of ILC3-secreted cytokines—IL17 and G-CSF, presented in violin plot, respectively. Each dot represents one animal, and the *p*-value is labelled (*n* = 6 mice). (**B**): Merged scan with an objective lens with a magnification power of 20 times (20×) demonstrating representative immunofluorescence staining for GATA3 (ILC2 nuclear marker, green), with DAPI (nuclear staining, blue) on WT and KO mouse colon (jelly-rolled, *n* = 3 mice stained). Scale bars = 50 µm. GATA3^+^ ILC2s presented at lamina propria were only observed in colons of KO mice.

**Figure 7 biomedicines-11-02734-f007:**
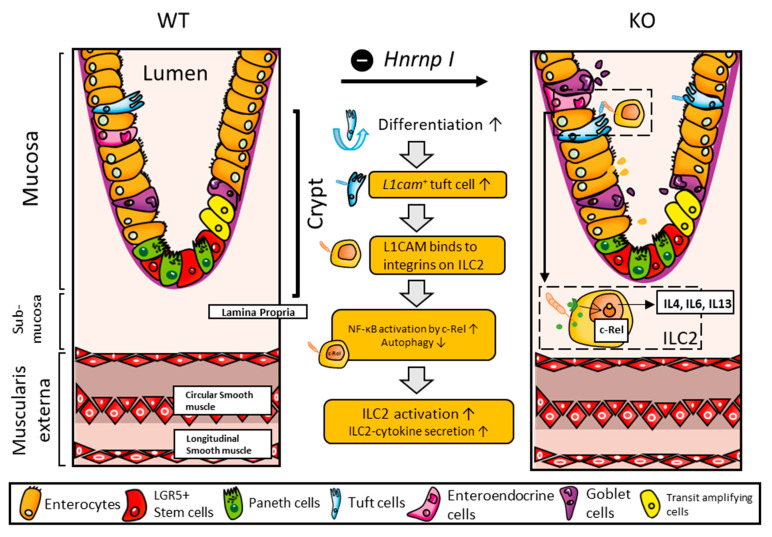
Schematic overview: intestinal epithelial cell-specific *Hnrnp I* knockout promotes colon inflammation, activating intestinal stem cell differentiation to more tuft cells. Tuft cells expressing high levels of *L1cam* communicate with ILC2 via integrin subunits (αv and β3), activating *c-Rel* dependent NF-κB signaling in ILC2s, promoting ILC2 activation and effector function by limiting autophagy while increasing the secretion of inflammatory cytokines (IL4, IL6, and IL13).

## Data Availability

Single-cell RNA sequencing raw data will be made available to other researchers on Gene Expression Omnibus (GEO, https://www.ncbi.nlm.nih.gov/geo/ (accessed on 1 October 2023)) once the article is accepted. Analytic R packages (Seurat, CellChat) used in the current study are available at https://satijalab.org/seurat/index.html (accessed on 20 June 2020) and http://www.cellchat.org/ (accessed on 28 June 2020). Analytic software used in the current study (10× Genomic Loupe Browser) is available at https://www.10xgenomics.com/products/loupe-browser (accessed on 1 June 2020).

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
