# Peer review of "Single-Cell RNA Sequencing (scRNA-seq) Identifies L1CAM as a Key Mediator between Epithelial Tuft Cell and Innate Lymphoid Cell in the Colon of Hnrnp I Knockout Mice"

_biomedicines, 2023, doi:10.3390/biomedicines11102734_

Round 1
Reviewer 1 Report
It is a very interesting and well-developed manuscript.
The topic is promising and relevant in the field.
The conclusion are consistent with the whole paper
The authors should implement some sections:
In particular, the limitations should be described as well as the clinical implications
In the introduction or discussion, it is necessary to introduce a comparison with other biomarkers
KRAS and BRAF mutations in serum exosomes from patients with colorectal cancer in a Chinese population. Oncol Lett. 2017 May;13(5):3608-3616. doi: 10.3892/ol.2017.5889
Mast Cells, microRNAs and Others: The Role of Translational Research on Colorectal Cancer in the Forthcoming Era of Precision Medicine. J Clin Med. 2020 Sep 3;9(9):2852. doi: 10.3390/jcm9092852
The authors should add also the proper checklist for the specific type of study (ARRIVE)
minor
Author Response
Please see the attachment "responses to reviewer 1".

Reviewer 2 Report
The manuscript by Xu and colleagues is focused on the scRNA-seq dataset generated using a murine model of intestinal inflammation. The model has been developed by the authors through specific KO of a gene encoding a protein belonging to the hNRNP family. On the one hand, this inflammation model has been previously characterized by the authors. On the other hand, the mechanism through which KO of HNRNPI leads to chronic intestinal inflammation is only in part characterized. Exploiting scRNA-seq data and in particular analysis of cell cell communication as inferred by scRNA-seq data, the authors suggest a possible route, which is specifically active only upon HNRNPI KO, for the activation of ILC2 cells by Tuft cells.
The data are clearly presented and data analysis is conducted rigorously. However I foresee two major weaknesses that the authors may want to address:
1) The data reported in Figure 3 A seem quite weak. It would be extremely important to show, apart from the color intensity, how small is the likelyhood of the inferred interaction by presenting some numbers. Furthermore, the authors should comment on whether other possible interaction were reported that were specific to KO (or WT) and compare those (if any) in terms of probability score to the one reported, possibly discussing why they chose to focus only on that particular interaction (in case more are suggested by CellChat)
2) Although the authors present a number of experiments which prove the interaction between DCLK1 positive cells and L1CAM, the data reported in section 3.3 suggest only a correlation between HNRNPI KO and the expression of ILC2 specific cytokines, but, in my opinion, do not really provide any convincing causal link between L1CAM and the NF-KB mediated inflammatory response which results in the expression of that set of cytokines. The authors should either perform a further experiment to validate their hypothesis about L1CAM role in this context or more explicitly state that their findings are coherent with their hypothesis but further experiments will be required to formally prove the role of L1CAM in this context.
Author Response
Please see the attachment "responses to reviewer 2"

Reviewer 3 Report
1. line 107: The experimental method section needs to specify the number of experimental animals in each group. Did the mice die during the feeding process?
2. line 246-251: Is this paragraph appropriate for the statistical methods section?
3. line 394: modified “20X magnification” to “200× magnification”. All figures need to be checked and modified.
4. The discussion part needs to be divided into several topics according to the main problems explained by your experiments in order to make it more organized.
5. The innovation of the article should be stated in an appropriate position.
Minor editing of English language required
Author Response
Please see the attachment "Responses to reviewer 3".
